# Efficient Neutrophil Activation Requires Two Simultaneous Activating Stimuli

**DOI:** 10.3390/ijms221810106

**Published:** 2021-09-18

**Authors:** Sanne Mol, Florianne M. J. Hafkamp, Laura Varela, Neena Simkhada, Esther W. Taanman-Kueter, Sander W. Tas, Marca H. M. Wauben, Tom Groot Kormelink, Esther C. de Jong

**Affiliations:** 1Department Experimental Immunology, Amsterdam UMC, Location AMC, 1105 AZ Amsterdam, The Netherlands; s.mol@amsterdamumc.nl (S.M.); f.m.hafkamp@amsterdamumc.nl (F.M.J.H.); n.simkhada@amsterdamumc.nl (N.S.); e.taanman@amsterdamumc.nl (E.W.T.-K.); s.w.tas@amsterdamumc.nl (S.W.T.); t.grootkormelink@amsterdamumc.nl (T.G.K.); 2Department Biomolecular Health Sciences, Faculty Veterinary Medicine, Utrecht University, 3508 TD Utrecht, The Netherlands; l.a.varelapinzon@uu.nl (L.V.); M.H.M.Wauben@uu.nl (M.H.M.W.); 3Amsterdam Rheumatology and Immunology Center, Department of Rheumatology and Clinical Immunology, Amsterdam University Medical Centers, University of Amsterdam, 1105 AZ Amsterdam, The Netherlands

**Keywords:** degranulation, mediator release, ROS production, NETosis, phagocytosis, extracellular vesicle release

## Abstract

Neutrophils are abundantly present in the synovium and synovial fluid of patients suffering from arthritis. Neutrophils can be activated by a multitude of stimuli and the current dogma states that this is a two-step process, consisting of a priming step followed by an activation step. Considering that neutrophil activation occurs in an inflammatory environment, where multiple stimuli are present, we argue that a two-step process is highly unlikely. Here, we indeed demonstrate that neutrophils require simultaneous ligation of two different receptors for efficient activation. We isolated human peripheral blood neutrophils and cultured them with various combinations of stimuli (GM-CSF, fMLF, TNF, and LPS). Next, we evaluated essential neutrophil functions, including degranulation and ROS production using flow cytometry, mediator release using ELISA, NETosis by a live cell imaging method, phagocytosis by imaging flow cytometry, and extracellular vesicle (EV) release quantified by high-resolution flow cytometry. Exposure of neutrophils to any combination of stimuli, but not to single stimuli, resulted in significant degranulation, and mediator and EV release. Furthermore, ROS production increased substantially by dual stimulation, yet appeared to be more dependent on the type of stimulation than on dual stimulation. Phagocytosis was induced to its maximum capacity by a single stimulus, while NETosis was not induced by any of the used physiological stimuli. Our data indicate that neutrophil activation is tightly regulated and requires activation by two simultaneous stimuli, which is largely independent of the combination of stimuli.

## 1. Introduction

Neutrophils are usually the first responders to an infection, and their primary role is killing invading pathogens [1]. In addition, neutrophils influence other immune cells and thereby can modulate both innate and adaptive immune responses [2,3]. Furthermore, neutrophils are present in high amounts in synovium and synovial fluid of arthritis patients, where they can have damaging effects in inflamed joints of these patients [4]. In peripheral blood, neutrophils are present in a resting state but in response to invading microbes, neutrophils migrate and become activated at the site of infection. Neutrophils employ a variety of mechanisms to eliminate pathogens and modulate the function of surrounding cells [5,6]. These mechanisms include degranulation, phagocytosis, reactive oxygen species (ROS) production, and the release of soluble mediators, neutrophil extracellular traps (NETs), and extracellular vesicles (EVs) [1,7,8,9]. Because of the cytotoxic nature of the majority of neutrophil-derived compounds, uncontrolled neutrophil activation can lead to tissue damage, for example in autoimmune diseases [6]. Indeed, neutrophils from rheumatoid arthritis (RA) and systemic lupus erythematosus (SLE) patients exhibit excessive ROS production and NETs contribute to development and disease activity [10]. Therefore, tight regulation of neutrophil migration and activation is important for optimal health.

The current dogma is that neutrophil activation is a two-step process: first neutrophils require pre-activation, also known as priming, which allows them to respond to an activating stimulus to become fully activated [11,12]. Such a process would be plausible if priming occurs by factors involved in migration (e.g., chemokine receptors or adhesion molecules), while activating factors comprise pathogen associated molecular patterns (PAMPs) and damage associate patterns (DAMPs). However, in the literature there is no clear distinction made between priming and activating stimuli. For example, LPS and fMLF are reported to act both as priming and activating stimuli [12,13,14,15,16,17,18,19], which is physiologically unrealistic, as both molecules are PAMPs. In contrast, GM-CSF and TNF are often described as priming stimuli [12,20,21], but these two cytokines are usually abundantly present at inflammatory sites [22]. Consequently, as neutrophils encounter pathogens that carry or induce many different inflammatory stimuli at the same time at the infection site, the true existence of a (spatio)temporal, two-step activation process may often be unlikely. Therefore, we hypothesized that the current designation of priming and activating stimulating agents is a rather artificial concept that is often not physiologically relevant. In this study, we used four different stimuli (GM-CSF, fMLF, TNF and LPS) to show that indeed dual stimulation is required to induce optimal neutrophil degranulation, cytokine release, ROS production and EV release. Consequently, we propose to dismiss the distinction between priming and activating stimuli and that efficient neutrophil activation is primarily a result of ligation of at least two activating receptors simultaneously, rather than a two-step priming and activating process.

## 2. Results

### 2.1. Dual Stimulation of Neutrophils Is Necessary for Efficient Degranulation

Full neutrophil activation is induced, according to literature, by subsequent exposure of a priming and activating stimulus. As certain stimuli are used both for priming and activating the stimulus, we questioned this system. Therefore, in order to investigate the effect of different stimulating agents on neutrophil activation, we measured degranulation, a well-known and essential antimicrobial mechanism of neutrophils. Neutrophils contain four different types of granules: azurophilic granules, specific granules, gelatinase granules, and secretory vesicles [23,24]. We analyzed CD63, CD66b, and CD16 membrane expression to determine the fusion of different granules with the plasma membrane upon activation. CD63 is present in azurophilic granules [23] and CD66b is present in specific gelatinase granules [23,24]. CD16 is expressed by resting neutrophils, but is also present in secretory vesicles [25], and is cleaved from the surface by the sheddase ADAM17 upon activation [26]. We defined fully degranulated neutrophils as CD16-CD63+ (Figure 1A) and the gating strategy is shown in Appendix A. These CD16-CD63+ neutrophils consistently displayed high CD66b expression (Figure 1B). We determined the CD16, CD63 and CD66b expression at 2 h after start of culture, since this culture period resulted in the detection of a clear activated phenotype (Appendix A) with limited cell death (Appendix A).

Neutrophils stimulated with either fMLF or TNF alone at increasing concentrations did induce, to different levels, degranulation in part of the neutrophils (Figure 1C). Additionally, the highest concentrations of GM-CSF significantly induced full degranulation, although this was with a very limited number cells (mean < 1.2%). Importantly, large differences were observed between individual donors, especially after incubation with fMLF and TNF (Figure 1C). For subsequent experiments, we selected suboptimal dosages that did not induce full degranulation in the majority of donors. Exposure of neutrophils to a combination of two stimuli consisting of either GM-CSF (50 U/mL), LPS (10 ng/mL), fMLF (1 µM), or TNF (1 ng/mL), resulted in a strong synergistic decrease of CD16 membrane expression (Figure 1D), and increase of CD63 and CD66b membrane expression (Figure 1E,F) compared to a single stimulus or no stimulus. This effect was observed with any combination of stimuli. Interestingly, the combination of two previously termed priming stimuli (GM-CSF and TNF) induced similar neutrophil degranulation as a combination of two activating stimuli (fMLF and LPS) or a more standard combination of a priming and an activating stimulus (GM-CSF and LPS). Although cell death was consistently higher upon double stimulation compared to single stimulation, this was always below 5% after 2 h of stimulation (shown for GM-CSF and LPS in Appendix A).

Another important feature of neutrophil activation is the loss of L-selectin (CD62L), which is important for neutrophil migration [27]. In contrast to degranulation, CD62L membrane expression is completely downregulated by neutrophils activated with any of the single and double stimuli used (Appendix A).

To enter the inflammatory site, neutrophils have to be exposed to chemotactic agents, like IL-8. To mimic this, we incubated neutrophils in the presence of IL-8 and/or LPS. We observe no ‘priming’ effect of IL-8, since degranulation was not induced in neutrophils stimulated with IL-8 and LPS compared to neutrophil stimulated with LPS alone (Appendix A).

Together, these data demonstrate that activation of neutrophils with more than one stimulus leads to a synergistic enhancement of neutrophil degranulation. Importantly, the choice and combination of previously termed priming and activating stimuli appears irrelevant to induce neutrophil degranulation and CD16 cleavage.

### 2.2. Dual Stimulation of Neutrophils Is Required for Optimal Mediator Release

Upon neutrophil activation, many different mediators are released into the extracellular environment, such as pre-stored antimicrobial peptides, or de novo synthesized cytokines, and chemokines [28,29]. We analyzed the release of neutrophil elastase (NE), a well-known pre-stored mediator in azurophilic (CD63 positive) granules. In line with the CD63 expression data described above, the release of NE was significantly increased by dual stimulation compared to single stimulation (Figure 2A), and significantly correlated (R = 0.76) with full degranulation (Figure 2B).

In contrast to NE, IL-8 is synthesized de novo upon activation and is regulated differently from expression and release of NE [28]. Consequently, IL-8 release may depend differently on single or dual stimulation. Single stimuli hardly induced any IL-8 secretion, while dual stimulation, independent of the combination of activating or priming stimuli, significantly enhanced IL-8 release (Figure 2C). Similarly for degranulation, we observed high inter-individual differences in IL-8 release, and although both degranulation and IL-8 were positively correlated (Figure 2D), this association was not straightforward. High IL-8 release always corresponded with high degranulation, while fully degranulated neutrophils did not always release high amounts of IL-8. We could not relate this discrepancy to individual differences, not to different stimulatory factors.

Collectively, similarly to degranulation, for optimal mediator release, a dual stimulation is essential, independent of the combination of stimuli used.

### 2.3. ROS Production Is Less Dependent on Dual Stimulation

Next, we investigated whether efficient ROS production is dependent on one or more activating stimuli or on a specific stimulus. Intracellular ROS production was determined by flow cytometry using the ROS indicator dihydrorhodamine-123 (123-DHR) (Figure 3A). ROS production was generally unaffected in neutrophils stimulated with either GM-CSF or LPS at increasing concentrations, although LPS induces ROS production in some donors. In contrast, neutrophils stimulated with fMLF or TNF clearly increased in ROS production at higher concentrations. Again, just as for neutrophil degranulation, we observed large differences in ROS production between individual donors after incubation with either LPS, fMLF, or TNF (Figure 3B,C). Overall, fMLF showed to be the most prominent inducer of ROS production. Neutrophils activated with two stimuli containing fMLF in combination with either LPS or TNF, but not GM-CSF, significantly further increased ROS production compared to single stimulated and unstimulated neutrophils. With other combinations of stimuli we observed a trend in increased in ROS production compared to single stimulated cells, but this was not significant nor seemed to be synergistic (Figure 3C). These data indicate that although dual stimulation induces strong ROS production, fMLF is an important stimulator of ROS production, especially when combined with LPS or TNF. ROS production is therefore more dependent on the type of stimulation than on dual stimulation.

### 2.4. Individual Stimuli or Combinations Do Not Induce NETosis

NETosis is a specific process employed by neutrophils in order to entrap and kill pathogens through the release of nuclear content to form a meshwork of chromatin, citrullinated histones and antimicrobial molecules [30,31]. Since neutrophil activation with GM-CSF and LPS induced profound degranulation, mediator release and ROS production, we first only assessed NETosis after stimulation with GM-CSF, LPS, or the combination of both. Unexpectedly, NETosis was not observed at any time point, whereas the non-physiological stimulus PMA induced clear NETosis (Figure 4A–C). After 4 h of stimulation with PMA, a clear presence of NETs was observed, which was less after 8 and 16 h of stimulation, probably due to the disappearance of DNA from neutrophils that underwent NETosis. In parallel, increasing numbers of dead cells were detected in response to PMA stimulation (data not shown). Although NETosis was not observed with GM-SCF/LPS stimulation, we also analyzed NETosis upon stimulation with fMLF or TNF, or any of the possible combinations of two stimuli. None of these stimuli could induce NETosis within 16 h of stimulation (data not shown). These results show that NETosis is not induced by single or combined stimulation with GM-CSF, LPS, fMLF, and/or TNF at concentrations that induce strong degranulation, mediator release, and ROS production.

### 2.5. Phagocytosis Is Independent of Dual Stimulation

Phagocytosis is another mechanism employed by neutrophils to engulf and eliminate pathogens. We investigated the effect of differential activation with single and double stimuli on phagocytosis using 1µm-sized FITC-labeled melamine beads. The proportion of neutrophils that engulfed FITC-labeled beads was quantified by flow cytometry (Figure 5A). Neutrophils stimulated with either single or double stimulation, using GM-CSF and LPS, demonstrated a significant increase in phagocytosis compared to unstimulated neutrophils, which was further increased by the addition of another stimulus (Figure 5B). To verify whether FITC-positive neutrophils detected by flow cytometry were positive as a result of bead uptake, and not due to aspecific binding to the extracellular plasma membrane, we performed ImageStream analyses. These analyses demonstrated no uptake when cells were kept at 4 °C, whereas the beads were efficiently taken up at 37 °C (Figure 5C). Moreover, with ImageStream analysis, we confirmed that FITC-positive cells were due to bead internalization, implying that the increased detection of FITC-positive neutrophils shown in Figure 5B was caused by phagocytosis. Collectively, these data indicate that phagocytosis is, in general, readily induced by a single stimulation and further increased by an additional stimulus.

### 2.6. Dual Stimulation of Neutrophils Increases the Release of EVs

EVs, small lipid bilayer-enclosed vesicles released by cells, are an important mode of communication between cells and are released in a controlled manner by many cell types, including neutrophils. Here we investigated whether EV release was influenced by dual stimulation. EV populations were isolated from culture supernatants of unstimulated, single, or dual stimulated neutrophils using GM-CSF and LPS and pelleted at 10,000 g (10 kg) or 100,000 g (100 kg), followed by sucrose density gradient separation. First, we determined with western blot analysis the presence of common EV proteins CD9 and CD63. For 10 kg EVs CD9 and CD63 signals were observed in the expected EV densities (1.18–1.22 g/mL and 1.10–1.16 g/mL) for both unstimulated and dual-stimulated neutrophils (Figure 6A). Furthermore, a CD63 signal, but not a CD9 signal, was observed in the 1.24 g/mL–1.27 g/mL density fraction of dual-stimulated neutrophils, which could be caused by massive degranulation (Figure 6A). For 100 kg EVs CD9 and CD63 signals were mainly observed in EV densities 1.10–1.16 g/mL for dual stimulated neutrophils and these signals were much lower for unstimulated neutrophils (Figure 6A and Appendix A). Importantly, dual stimulation strongly increased the signal of CD9 and CD63 of 100k EVs in 1.10–1.16 density fraction. To analyze the amount of EV release by neutrophils, EVs pelleted at 10 k or 100 kg were labeled with a lipophilic dye (PKH67) followed by sucrose density gradient separation and subsequently analyzed using fluorescence triggered single EV flow cytometric analysis [32]. The scatter plots of time-based measurements of individual fluorescently labeled EVs in fractions 1.18–1.22 g/mL and 1.10–1.16 g/mL show clear differences in scatter profiles between 10 k and 100 k EVs in all conditions from unstimulated, single- and dual-stimulated neutrophils, indicating differences in EV subsets. This corroborates the differences in CD9 and CD63 distribution between these fractions as observed in the western blot. Furthermore, a strong increase in the amount of EVs in the 1.10–1.16 g/mL density fraction was observed after dual stimulation which match the Western blot data (Figure 6B). Quantitative analysis shows that activation of neutrophils by dual stimulation significantly increased the release of EVs and especially of the 100 kg EVs in density fraction 1.10–1.16 g/mL (Figure 6C). In line with this observation, the number of 100 k EVs released by neutrophils moderately correlated with neutrophil degranulation (Appendix A). Although the amount of cell death was always low, a moderate correlation between 10 kg EVs present in 1.10–1.16 g/mL density fraction and dead cells was observed. There was no correlation observed between 10 k EVs present in 1.18–1.22 g/mL and 100 kg EV number and percentage of dead cells (data not shown), indicating that most 100 kg EVs detected are released by viable cells and were neither released as a result of apoptosis nor other types of cell death. Taken together, these data indicate that dual stimulated neutrophils do release more EVs than unstimulated or single-stimulated neutrophils which is most prominent for 100 kg EVs in density fraction 1.10–1.16 g/mL.

## 3. Discussion

In the current study we provide evidence that peripheral blood neutrophils from healthy donors require two simultaneous stimuli for optimal activation resulting in degranulation, mediator release and EV release. In contrast, ROS production is more dependent on the type of stimulation than on dual stimulation, while phagocytosis appears already close to its maximum capacity when only one type of stimulus is present. Surprisingly, we found that the neutrophil function NETosis was not induced by any of the used physiological stimuli or combinations thereof, but could only be induced by the artificial stimulus PMA. Moreover, our data suggest that the current use of priming and activating agents is often not necessarily physiologically relevant. Instead, we demonstrated that neutrophils require activation by two different stimuli which is independent of the combination of stimuli. Any combination of GM-CSF, TNF (both generally considered priming stimuli), LPS, or fMLF (both generally considered activating stimuli) are very well capable of inducing various aspects of neutrophil activation. This contradicts the current dogma stating that neutrophil activation is a two-step process of priming followed by activation. Certain single stimuli; however, already induce neutrophil activation to some extent. At higher concentrations (generally ≥1 ng/mL) TNF enhances degranulation and increases ROS production compared to unstimulated neutrophils. Importantly, our results indicate that degranulation or IL-8 release is strongly increased by the combination with another stimulus.

Quite unexpectedly, we observed a large variation in neutrophil degranulation and ROS production between donors, both after applying a single (except for GM-CSF) and a double stimulus. Neutrophils from some donors did not react at all to a single stimulation, while neutrophils from other donors showed some degree of activation. Although at present we cannot explain these inter-individual differences, the high donor-to-donor variation may reflect the effectiveness of neutrophils to react to certain microbes. On the other hand, it may mirror the propensity of individuals to disorders associated with neutrophil activation-induced damage. It has been previously described that immunosenescence could play a role, because neutrophils from aged donors (above 60 years of age) can respond differently [33]. However, this cannot account for the observed donor-to-donor variation, since all of our donors were below 60 years of age. Furthermore, we observed no correlation between age and neutrophil degranulation or ROS production.

These observations question the distinction between priming and activating stimuli. TNF is a stimulus that is classically linked to priming whereas it can induce activation of neutrophils itself to some extent. In contrast, LPS is mostly designated as an activating stimulus but not, or only moderately induces any neutrophil activation when presented as a single stimulus.

For this study we isolated neutrophils from peripheral blood using Lymphoprep density centrifugation directly after drawing blood. Many isolation methods have been indicated to increase neutrophil activation [34]. In contrast, our isolation methods seem to have a limited effect, as unstimulated neutrophils truly resemble unstimulated neutrophils. Furthermore, it has been suggested that chemo attraction may influence neutrophil activation. However, we did not observe any effect on degranulation of the potent chemoattractant IL-8, alone or in combination with LPS. The same is true that other stimuli may have a different effect on neutrophil activation. In this study, we only use four different stimuli for neutrophil activation. These are four widely used stimuli in terms of neutrophil activation.

In contrast to degranulation, mediator release, ROS production, EV release and to some extent phagocytosis, which were significantly enhanced by ligation of two types of receptors, no NETosis was observed with any combination of stimuli tested. The formation of NETs, consisting of nuclear chromatin with contents of granules and antimicrobial molecules, has been identified as an important mechanism of neutrophils to trap and disarm invading microbes. Moreover, NETosis has been associated with various chronic inflammatory and infectious diseases including SLE [10,35], RA [36], and COVID-19 [37]. We only observed NETosis when neutrophils were stimulated by PMA, a very robust but artificial stimulus, indicating that the assay that we set up to measure NET formation is valid. Some earlier studies have observed NETosis in response to TNF and LPS [13,38], whereas others could not detect NETosis with these stimuli [39]. A reason for these differences could be the difference in LPS that was used, or the culture condition. Previous research has shown that neutrophils can selectively release NETs when stimulated with LPS from different bacterial sources under serum-free conditions [40]. Another factor that may play a role is the use of FCS as it has recently been reported that serum and serum albumin inhibit LPS-induced in vitro formation of NETs [41]. No serum or only 0.5% serum was used in NETosis assays, where a response was observed to LPS [13,38], but this setup resulted in rapid cell death in our assay. Since we could detect PMA-induced NETosis in medium containing 10% FCS, we find this a less likely explanation for the absence of NETosis. GM-CSF and TNF are cytokines that are abundant in disorders associated with NETs and frequently linked to disease pathogenesis [42,43]. Circulating TNF levels were elevated in SLE patients versus controls and TNF was proposed as biomarker for SLE disease activity [44]. Therefore, it is surprising that these cytokines do not induce NETosis in our study. This may indicate that either the induction of NETosis is induced by a different set of stimuli, via a different mechanism, or at other concentrations of the stimuli than those used in this study. Another more valid explanation could be that induction of NETosis requires a more complex activation system such as multiple stimuli that are present in microbial stimulation. The latter mechanism would be physiologically reasonable as the formation of NETs has been associated with detrimental tissue damage [6,45,46] and therefore requires tight regulation with various safety valves.

Here we show that the release of EVs by neutrophils also is enhanced by a dual stimulation. Nevertheless, single stimulation by GM-CSF or LPS also induced EV release. We recently showed that LPS-induced neutrophils increase in 100 k EV release, and that this is correlated with degree of degranulation [47], a finding that is supported by data presented in this study. However, neutrophils stimulated with GM-CSF showed little to no increase in degranulation was induced, while there was an increased release of small EVs. This indicates that increased EV release can occur independently of primary or secondary granule release. The increase of EVS was most pronounced in EVs pelleted at 100 kg. Although we did not observe clear differences in the mean buoyant density between these two EV types, the light scattering signals do show differences between the 10 kg and 100 kg EVs, which could indicate differences in size and/or composition. Since molecular cargo selection of EVs is not a random process, but rather regulated via distinct and regulated processes that are dependent on the type and state of the cell, the type of stimuli that were received, and the subcellular origin of EVs [48,49]. For mast cells for example, we previously observed clear differences in the buoyant density, light scattering, and CD9 and CD63 content of EVs released from unstimulated and activated cells [50]. Moreover, it has been shown that neutrophil-derived EVs indeed have a distinct protein composition, and have antimicrobial capacities [51,52]. It remains to be elucidated whether these composition differences also exist between EVs of single or double stimulated neutrophils.

The observation that a single stimulus is insufficient to fully activate a cell, and to induce a full executive program is not novel. For instance, abundant IL-12 production by dendritic cells is only induced by a double stimulus, e.g., a PAMP in combination with IFN-γ [53]. IL-12 is an important activator of Th1 cells that via the production of IFN-γ activate macrophages, which may cause massive tissue damage when over activated. Furthermore, for the production and secretion of high levels of IL-23 and IL-1β, important cytokines for Th17 cell development, a PAMP in combination with Fc-receptor ligation on dendritic cells is necessary [54]. Additionally, IL-17-producing Th17 cells have been shown to be potentially detrimental and they are associated with various chronic inflammatory disorders. Nevertheless, they are crucial in the fight against bacterial and fungal infections [55]. These examples underline that important processes that are potentially detrimental to the host when overactivated, are often regulated at multiple levels. This could be a physiological explanation as to why neutrophils require more than one stimulus in order to start releasing a plethora of harmful contents.

In this study, we show that full neutrophil activation is tightly regulated and requires the ligation of at least two different types of receptors. Moreover, there is no need for a spatio-temporal separation of these stimulatory actions, as neither would be the case where a pathogenic microbe enters the body resulting in rapid activation of neutrophils. Infection or inflammation results in the rapid release of inflammatory associated molecules such as TNF and GM-CSF. These cytokines are also highly active in arthritis patients. As neutrophils are exposed these and other stimuli concomitantly, full activation of neutrophils will happen rapidly, resulting in the killing and elimination of invading microbes by one or more of the effective mechanisms that neutrophils possess or leading to excessive damage as observed in arthritis.

## 4. Materials and Methods

### 4.1. Neutrophil Isolation

Blood was collected from healthy volunteer donors after informed consent into sodium heparin tubes (Greiner Bio-One, Alphen a/d Rijn, The Netherlands). Blood was diluted with Hanks balanced salt solution (HBSS, Sigma-Aldrich Inc., St. Louis, MO, USA) and granulocytes and erythrocytes were separated from peripheral blood mononuclear cells (PBMCs) by density gradient centrifugation on Lymphoprep (d = 1.077 ± 0.001 g/mL; Axis-Shield, Oslo, Norway). Erythrocytes were lysed in ice-cold erythrocyte lysis buffer (containing 0.155 M NH_4_Cl (Sigma-Aldrich, Inc., St. Louis, MO, USA), 1 mM KHCO_3_ and 80 µM EDTA (both Merck KGaA, Darmstadt, Germany), dissolved in sterile water, pH 7.3) for 10 min on ice. Subsequently, neutrophils were cleared from remaining erythrocytes with a second lysis step for 5 min on ice, and washed twice in PBS. Neutrophils were then resuspended in IMDM (Gibco; Thermo Fischer Scientific Inc, Waltham, MA, USA) supplemented with 10% heat inactivated (HI) fetal bovine serum (FBS; Hyclone; Thermo Fischer Scientific Inc, Waltham, MA, USA) and gentamycin (86 µg/mL; Duchefa Biochemie B.V., Haarlem, The Netherlands) and used immediately. Neutrophil purity was analyzed by flow cytometry and was always >97%.

### 4.2. Neutrophil Culture, Stimulation and Flow Cytometric Analysis

Neutrophils were seeded at a density of 0.4 × 10^6^ cells/mL in 250 µL in a flat bottom 96-well plate (Costar, Corning Inc., Corning, NY, USA) in IMDM medium containing 10% HI-FBS and gentamycin. subsequently, neutrophils were cultured for 1, 2, or 24 h at 37 °C in the absence or presence of the following reagents: granulocyte-macrophage colony-stimulating factor (GM-CSF) (Schering-Plough B.V., Brussels, Belgium, catalog no. PSR 99M0408), N-formyl-methionyl-leucyl-phenylalanine (fMLF) (Sigma, catalog no. F3506), lipopolysaccharide (LPS) (Sigma-Aldrich, catalog no. L3024, from *Escherichia Coli.* O111:B4), and tumor necrosis factor (TNF) (Miltenyi Biotec, catalog no. 130-094-022). After 2 or 24 h culture, supernatants were collected for the analysis of neutrophil elastase and IL-8, respectively. For flow cytometric analysis of ROS production, neutrophils were cultured and stimulated for 1 h at 37 °C in the presence of 25 µM 123-dihydrorhodamine (123-DHR; Marker Gene Technologies, Eugene, OR, USA). For the flow cytometric analysis of CD63, CD66b, CD16, and CD62L, neutrophils were stimulated for 2 h. After stimulation, cells were harvested and washed twice in cold PBA (PBS-0.5% *w*/*v* BSA-0.05% *w*/*v* azide), followed by antibody labeling in PBA. Degranulation (CD63 and CD66b), CD16 and CD62L expression, and cell viability (PI) were determined using flow cytometry after 2 h of culture. The following antibodies were used: αCD15-FITC (1:100; HI98), αCD16-PECy7 (1:1000; 3G8), αCD62L-APCCy7 (1:25; Greg-56), αCD63-APC (1:100; H5C6), αCD66b-PE (1:100; G10F5), (all Biolegend, San Diego, CA, USA). DAPI dihydrochloride (DAPI; 20 µM) or propidium iodide (PI; 500 ng/mL) (both from Sigma-Aldrich) were used to determine cell viability. A total of 10,000 cells were acquired in the live gate on a FACSCanto (BD Biosciences, San Jose, CA, USA) and further analyzed using FlowJo software (BD Biosciences, San Jose, CA, USA).

### 4.3. IL-8 ELISA

IL-8 concentrations were determined in culture supernatants collected after 24 h of culture using an IL-8 ELISA (Invitrogen Life Technologies, Breda, The Netherlands). In brief, flat-bottom EIA/RIA 96-well plates (Costar, Corning Inc.) were coated at 4 °C overnight with aniti-IL-8 antibody (1:1000, 893A6G8, Invitrogen, Carlsbad, CA, USA) diluted in carbonate buffer (0.5M, pH 9.6). Plates were washed with PT (PBS-0.1% *v*/*v* TWEEN 20) and blocked with PTB (PT-1% *w*/*v* bovine serum albumin) at 37 °C for 1 h. Next, wells were incubated with culture supernatants (commonly diluted 10 times in PT) or recombinant IL-8 for 1 h, washed thrice with PT, and incubated for 1 h with IL-8 detection antibody (1:1000, 790A28G2, Invitrogen) diluted in PTB. Subsequently, wells were washed thrice and incubated for 45 min with strep-poly HRP (1:10,000, M2032, Sanquin, Amsterdam, The Netherlands) diluted in PT containing 2% protifar (Nutricia, Utrecht, The Netherlands). Finally, plates were washed five times and developed with 3,3′,5,5′-tetramethylbenzidine (TMB, Merck, Darmstadt, Germany). The reaction was stopped by adding an equal amount of 1M H2SO4. Absorbance was measured at 450 nm with reference at 655 nm by using a VersaMax microplate reader (Molecular Devices, Silicon Valley, CA, USA).

### 4.4. Neutrophil Elastase ELISA

Neutrophil elastase (NE) concentration was analyzed in culture supernatants collected after 2 h of cell culture as described in Souwer et al. [3]. Briefly, flat-bottom EIA/RIA 96-well plates were coated overnight with polyclonal rabbit IgG directed against NE (1.5 ng/mL, Sanquin) at 4 °C. Plates were washed with PT thrice before adding culture supernatants, usually diluted 250 times in PTG (PT-0.2% *v*/*w* gelatin), and incubated for 1 h. Next, plates were washed with PT and incubated for 1 h with biotinylated rabbit anti-human elastase diluted in PTG (1 ng/mL, Sanquin blood Supply). Subsequently, after a washing step, plates were incubated for 30 min with streptavidin-peroxidase diluted 1:1000 in PTG (Amersham Life Science, Buckinghamshire, UK). Finally, the ELISA was completed using TMB as a substrate and measured as described above.

### 4.5. NETosis Assay

NET formation (NETosis) was analyzed using an Incucyte S3 Live-Cell Analysis System (Essen BioScience, Ann Arbor, MI, USA) and a previously described IncuCyte^®^ NETosis assay [56]. Briefly, neutrophils were seeded at a density of 1.0 × 10^5^ cells/mL in 200 µL in a 96-well IncuCyte^®^ Imagelock plate (Essen BioScience, Ann Arbor, MI, USA) in IMDM medium containing 10% HI-FBS and gentamycin, and incubated with the above indicated stimuli, or with 100 ng/mL PMA (a well-known NET-inducing agent) for 16 h. The cell-impermeant nucleic acid binding dye YOYO™-3 Iodide (Invitrogen, Carlsbad, CA, USA) was present at 400 nM in the medium to stain free available DNA. Neutrophils were imaged every 15 min using phase contrast and red fluorescent exposure channels present in the IncuCyte, using a 20× dry objective lens. Data were analyzed using the IncuCyte Basic Software (Essen BioScience, Ann Arbor, MI, USA). Neutrophils were identified by phase contrast, objects smaller than 100 µm^2^ were excluded as cells. Neutrophils that underwent apoptosis were identified by red staining of the cell, neutrophils undergoing NETosis were identified by red staining in the cell and red staining visibly crossing the cell membrane, this was marked by red fluorescent objects larger than 400 µm^2^ in area. For the red fluorescent channel, edge sensitivity was set to 0 and hole fill was set to 100 µm^2^. The TopHat method and the edge split tool were used for background correction and for accurate quantification of individual cells. NETosis was quantified by the total signal of red fluorescent area of objects > 400 µm^2^ per well.

### 4.6. Phagocytosis Assay

Neutrophils were seeded at a density of 1 × 10^6^ cells/mL in 1 mL in a flat bottom 24-well plate in IMDM medium containing 10% HI-FBS and gentamycin, and stimulated with the above indicated stimuli for 2 h at 37 °C or at 4 °C in the presence of 0.0003% *w*/*v* FITC-marked microparticles based on melamine resin (FITC-beads; Sigma-Aldrich). After stimulation, neutrophils were harvested, washed twice in cold PBA buffer, and labeled in PBA buffer using the following antibodies: αCD16-PECy7 (1:1000; 3G8), αCD63-APC (1:100; H5C6), αCD66b-PE (1:100; G10F5) (all Biolegend). Analysis was done using flow cytometry and using advanced imaging flow cytometry. For this latter analysis, a total of 100,000 cells were acquired with an Amnis ImageStream (Luminex Corporation, Austin, TX, USA) and further analyzed using IDEAS software (Amnis, Seattle, WA, USA).

### 4.7. EV Isolation and Fluorescent Labeling

To prepare EV-depleted FBS, 30% FBS in IMDM was ultracentrifuged for 16 h at 100,000× g in an SW32 rotor (Beckman Coulter, Fullerton, CA, USA). For isolation of EVs, 6 × 10^6^ neutrophils were cultured per stimulation condition for 2 h in 24-well plates (Costar, Corning), 1 × 10^6^ neutrophils per well in 1 mL IMDM medium containing 1% EV-depleted FBS. After 2 h, culture supernatants were collected gently to minimize co-isolation of neutrophils, and sequentially subjected to differential centrifugation and floatation into sucrose gradients as described previously [32]. Briefly, supernatants were centrifuged twice at 200× *g* for 10 min, and twice at 500× *g* for 10 min. Subsequently, EVs were pelleted by ultracentrifugation of the 500× *g* supernatant for 30 min at 10,000× *g* (10 kg EVs) using an SW40 rotor (Beckman Coulter), and by ultracentrifugation of the collected 10,000× *g* supernatant for 65 min at 100,000× *g* (100 kg EVs) using an SW40 rotor in a Beckman Coulter ultracentrifuge at 4 °C. EV-containing pellets were resuspended in 20 µL PBS containing 0.2% EV-depleted bovine serum albumin (BSA) and labeled with 7.5 mM PKH67 (Sigma, St. Louis, MO, USA) in 180 µL diluent C. The reaction was stopped by adding 100 µL of IMDM with 10% EV-depleted FBS. PKH67-labeled pellets were mixed with 1.5 mL 2.5 M sucrose and a linear sucrose gradient (2.0–0.4 M sucrose in PBS) was added on top in a SW40 rotor. Gradients were centrifuged at 192,000× *g* (average) for 16–17 h at 4 °C, after which 1 mL gradient fractions were collected by pipetting from the top of the tube. Densities of all fractions were determined by refractometry.

### 4.8. Western Blotting

Gradient sucrose fractions from EV isolation 1–3 (1.27 g/mL–1.24 g/mL), 4–6 (1.22 g/mL–1.18 g/mL), 7–10 (1.16 g/mL–1.10 g/mL), and 11–12 (1.08 g/mL–1.06 g/mL) were pooled for protein identification by Western blotting and diluted in PBS in SW40 tubes (Beckman Coulter, Fullerton, CA, USA). EVs were pelleted at 125,755 g for 70 min at 4 degrees. Pellets were resuspended in 30 µL non-reducing SDS-PAGE sample buffer (0.25M Tris/HCl, pH 6.8, 40% *v*/*v* glycerol, 8% *w*/*v* SDS, 0.01% *w*/*v* bromophenol blue), heated at 100 degrees, run on pre-cast gel (Criterion TGX Gels, BioRad, Hercules, CA, USA) and transferred onto a 0.2 µm polyvinyldidene difluoride membrane. After blocking for 1 h in blocking buffer (0.2% Fish skin gelatin +0.1% Tween-20), blots were incubated overnight at 4 °C with primary antibodies against CD9 (clone HI9, Biolegend, San Diego, CA, USA; dilution 1:1000) and CD63 (clone H5C6, BD Bioscience, dilution 1:500,) in blocking buffer. Washed three times in PBS-tween and incubated for 1 h with HRP-coupled secondary antibody (Jackson ImmunoResearch, PA, USA; dilution 1:10,000). Blots were washed again three times in PBS-Tween, followed by two washes in plain PBS and incubation with ECL solution (ThermoScientific, SuperSignal West Dura Extended Duration Substrat, cat. 34075). Blots were analyzed using BioRad Chemidoc imager (BioRad, Hercules, CA, USA) and Rad Image Lab V5.1 software (BioRAD).

### 4.9. Neutrophil EV Quantification by High-Resolution Flow Cytometry

High-resolution flow cytometric analysis of PKH67-labeled EVs was performed using an optimized jet-in-air-based BD influx flow cytometer that was dedicated and optimized for detection of submicron-sized particles (BD Biosciences, San Jose, CA, USA). Detailed descriptions of both the hardware adaptions and methods used were previously described in detail [32,57]. Forward scatter was detected with a collection angle of 15–25° (reduced wide-angle forward scatter (rw-FSC)). Briefly, fluorescence threshold triggering was applied to distinguish PKH67-labeled EVs from non-fluorescent events. PKH67 was excited with a 488 nm laser (Sapphire, Coherent 200 mW) and fluorescence was collected through a 530/40 bandpass filter. Prior to EV-measurements fluorescent polystyrene 100 and 200 nm yellow-green (505/515) FluoSphere beads (Invitrogen, Carlsbad, CA, USA) were measured to calibrate the fluorescence and rw-FSC settings in order to minimize day to day variations. The instrument was aligned until predefined MFI and scatter intensities were reached with the smallest possible coefficient of variation for rw-FSC, SSC and fluorescence. After optimal alignment, PMT settings required no or minimal day to day adjustment. For EVs a fluorescence threshold was set allowing an event rate of 10–20 events/second while acquiring a clean PBS sample. For quantitative analysis of PKH67-labeled EVs sucrose gradient fraction between 1.06 g/mL and 1.20 g/mL were diluted at least 20× in PBS prior to analysis to keep the event rate < 10,000/s to avoid coincident particle detection and occurrence of swarm [58]. Upon loading on the influx, the sample was boosted into the flow cytometer until events appeared, after which the system was allowed to stabilize for 30 s. Measurements were performed by a fixed 30 s time. Scatter and EVs count data were analyzed using FlowJo software (BD Biosciences).

### 4.10. Statistical Analysis

Data are expressed as mean ± SD. Statistical analysis was done in GraphPad Prism version 8.3.0 for Windows by using statistical tests, depending on the experimental data. For multiple comparisons *p*-values were calculated on selected pairs either using a one-way ANOVA or mixed-effects analysis with Tukey post-test correction or using Friedman tests with Dunn post-test correction. For single comparisons *p*-values were calculated using paired *t*-tests. *p*-values < 0.05 were considered as statistically significant.

## Figures and Tables

**Figure 1 ijms-22-10106-f001:**
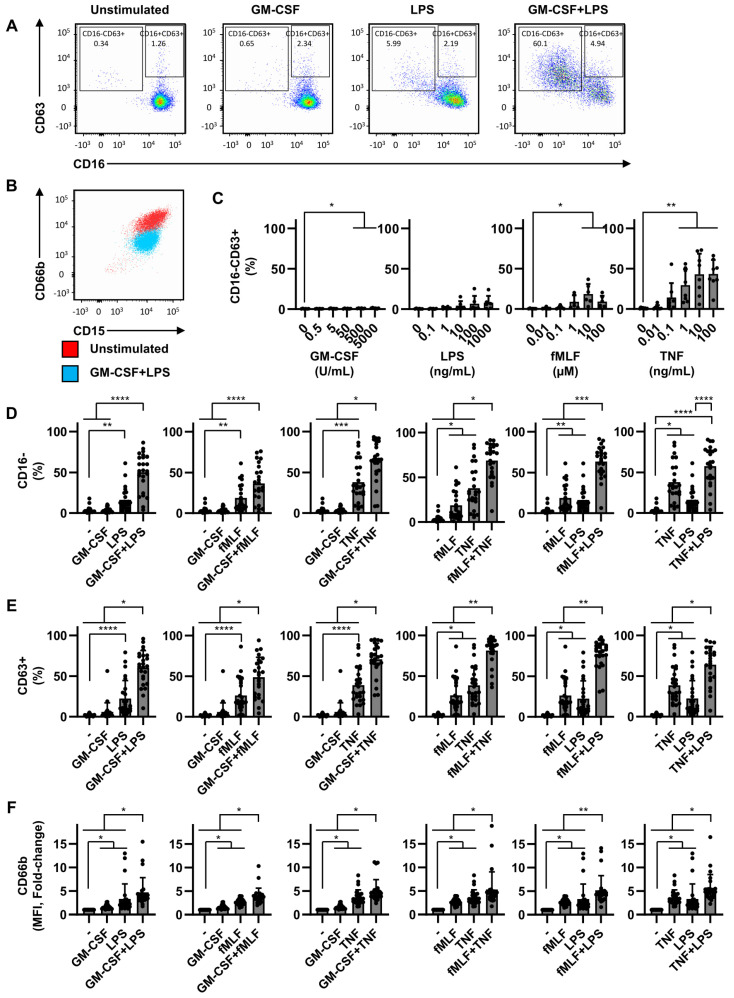
Dual stimulation of neutrophils is required for efficient degranulation. Neutrophils were cultured for 2 h in the absence or presence of different stimuli used at fixed concentrations (GM-CSF: 50 U/mL, LPS: 10 ng/mL, fMLF: 1 µM, and/or TNF: 1 ng/mL), (**A**,**B**,**D**–**F**) or increasing concentrations (**C**). (**A**) Flow cytometry plot demonstrating gating strategy to determine neutrophil degranulation: CD16-CD63+, or azurophilic degranulation: CD63+ (CD16-CD63+ and CD16+CD63+). (**B**) Flow cytometry plot demonstrating that neutrophils expressing CD16-CD63+ are consistently high in degranulation of specific and gelatinase granules (CD66b). (**C**) Effects of increasing concentrations of the different stimuli on neutrophil degranulation displayed as percentages of CD16-CD63+ neutrophils. (**D**) Effect of single and double stimulation on degranulation as measured by percentages of CD16- neutrophils. (**E**) Effect of single and double stimulation on degranulation as measured by percentages of CD63+ neutrophils. (**F**) Effect of single and double stimulation on specific and gelatinase granule degranulation as measured by changes in mean fluorescent intensity (MFI) of CD66b, expressed in fold change compared to unstimulated. Data are representative of 6 (**C**) and 25 (**D**,**E**,**F**) independent experiments, and are presented as mean ± SD. * *p* < 0.05, ** *p* < 0.01, *** *p* < 0.001, and **** *p* < 0.0001. The *p*-values were calculated using a paired *t*-test (**C**) and Friedman test with Dunn’s correction.

**Figure 2 ijms-22-10106-f002:**
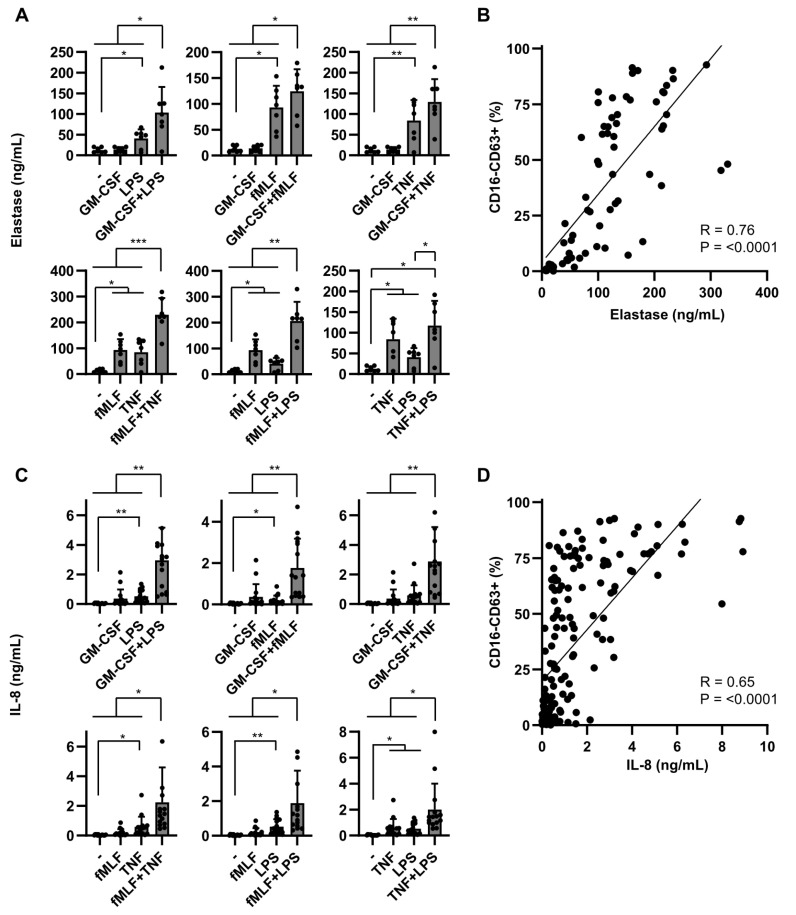
Dual stimulation of neutrophils is required for optimal mediator release. Neutrophils were cultured for either 2 or 24 h in the absence of presence of different stimuli (GM-CSF (50 U/mL), fMLF (1 µM), TNF (1 ng/mL) and/or LPS (10 ng/mL). (**A**) NE was measured in culture supernatants collected 2 h after activation (*n* = 7). (**B**) Correlation between NE release and degranulation. Neutrophil degranulation was assessed by flow cytometry as shown in Figure 1. Data consist of 7 independent experiments, with multiple conditions in each experiment. (**C**) IL-8 was measured in culture supernatants collected 24 h after activation (*n* = 11). (**D**) Correlation between IL-8 release and degranulation. Neutrophil degranulation was assessed by flow cytometry as shown in Figure 1. Data consist of 11 independent experiments, with multiple conditions in each experiment. (**A**,**C**) Data are presented as means ± SD, * *p* < 0.05, ** *p* < 0.01, *** *p* < 0.001, one-way ANOVA with Tukey’s correction. (**B**,**D**) Linear regression was applied to determine R.

**Figure 3 ijms-22-10106-f003:**
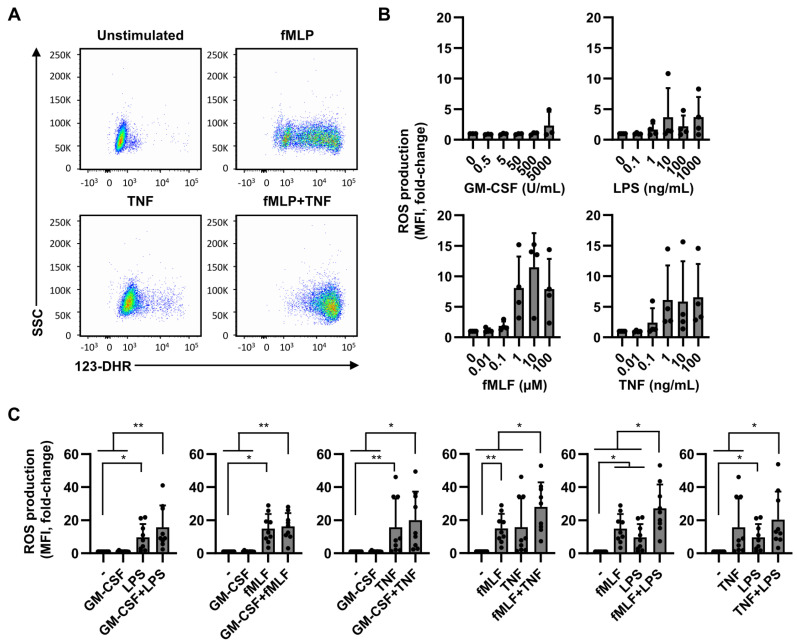
Effects of neutrophil activation by single or double stimuli on ROS production. Neutrophils were cultured in the presence of 123-DHR and in the absence or presence of GM-CSF (50 U/mL), fMLF (1 µM), TNF (1 ng/mL), and/or LPS (10 ng/mL). After 1 h of culture ROS production was analyzed using flow cytometry. (**A**) Representative dot plots showing the influence of neutrophil stimulation on ROS production. (**B**) Effects of increasing concentrations of the different stimuli on ROS production (*n* = 4). (**C**) Effect of single and double stimulation on ROS production. Total ROS production is expressed as fold change on MFI of 123-DHR compared to unstimulated. Data are presented as mean ± SD. * *p* < 0.05, and ** *p* < 0.01. The *p*-values were calculated using a paired *t*-test (**B**) and Friedman test with Dunn’s correction (**C**).

**Figure 4 ijms-22-10106-f004:**
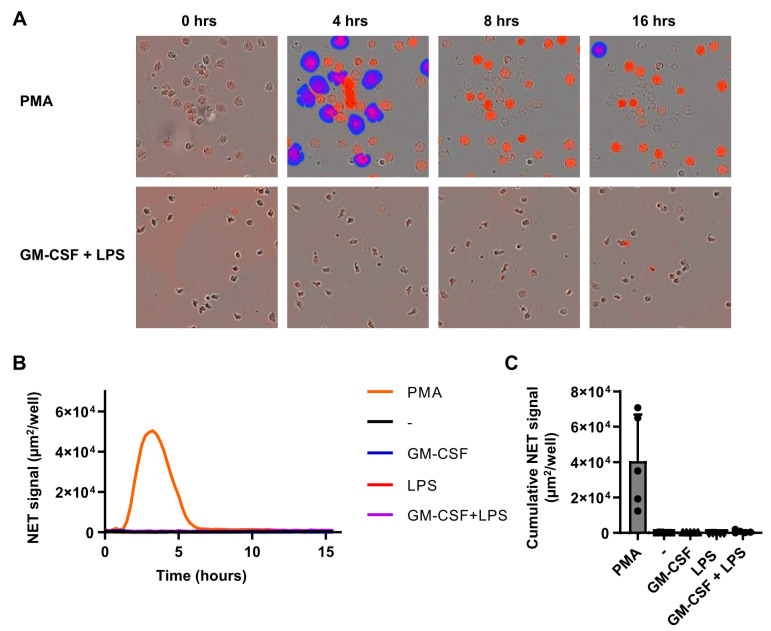
Physiological stimuli do not induce NETosis. Neutrophils were cultured for 16 h in the presence of a cell impermeable fluorescent DNA-binding dye in the absence or presence of GM-CSF (50 U/mL) and/or LPS (10 ng/mL), or PMA (100 µg/mL). Fluorescence was measured every 15 min to determine NET formation and cell death. (**A**) Overlays of phase contrast and fluorescence images showing accessible DNA in red, and extracellular DNA (>400 µm^2^) in blue (NETs). Images are representative of 5 independent experiments. (**B**) NETosis was determined as fluorescence signal area in µm^2^ per well. Only areas larger than 400 µm^2^ were used for calculations (*n* = 5). (**C**) NETosis expressed as fluorescence area after 4 h of stimulation (*n* = 5). Data are presented as mean ± SD.

**Figure 5 ijms-22-10106-f005:**
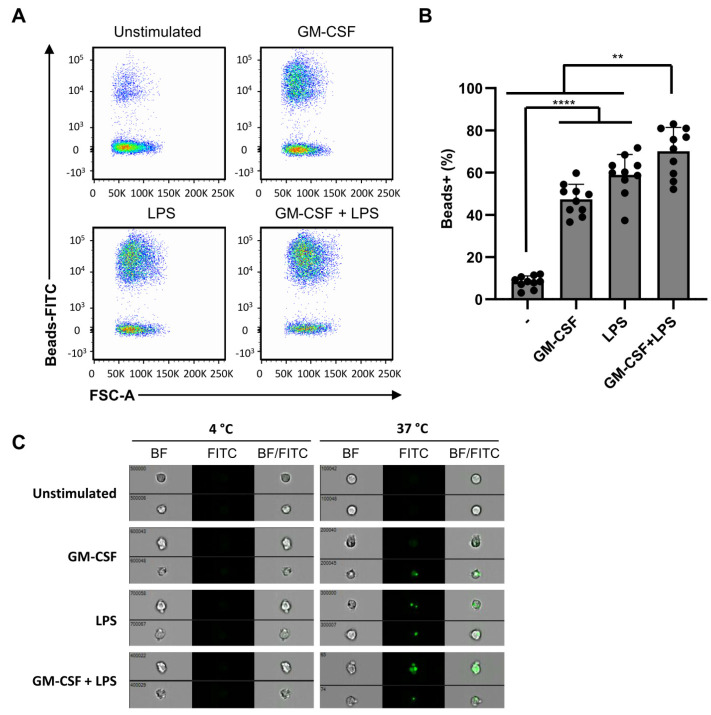
Phagocytosis is independent of dual stimulation. Neutrophils were stimulated for 2 h in the presence of FITC-labeled melamine beads and in the absence or presence of GM-CSF (50 U/mL), fMLF (1 µM), TNF (1 ng/mL), or LPS (10 ng/mL). (**A**) Phagocytosis was analyzed using flow cytometry. (**B**) Phagocytosis, determined by flow cytometry as percentages of FITC positive neutrophils (*n* = 10). (**C**) Imagestream microscopy images of neutrophils stimulated at 37 °C or 4 °C with GM-CSF (50 U/mL) and/or LPS (10 ng/mL) in the presence of FITC-labeled melamine beads. Location of phagocytosed beads is shown in green (left), and overlay of phase contrast neutrophil (right). Data are presented as mean ± SD. Asterisks indicate significant differences: ** *p* < 0.01 and **** *p* < 0.0001, one-way ANOVA with Tukey’s correction.

**Figure 6 ijms-22-10106-f006:**
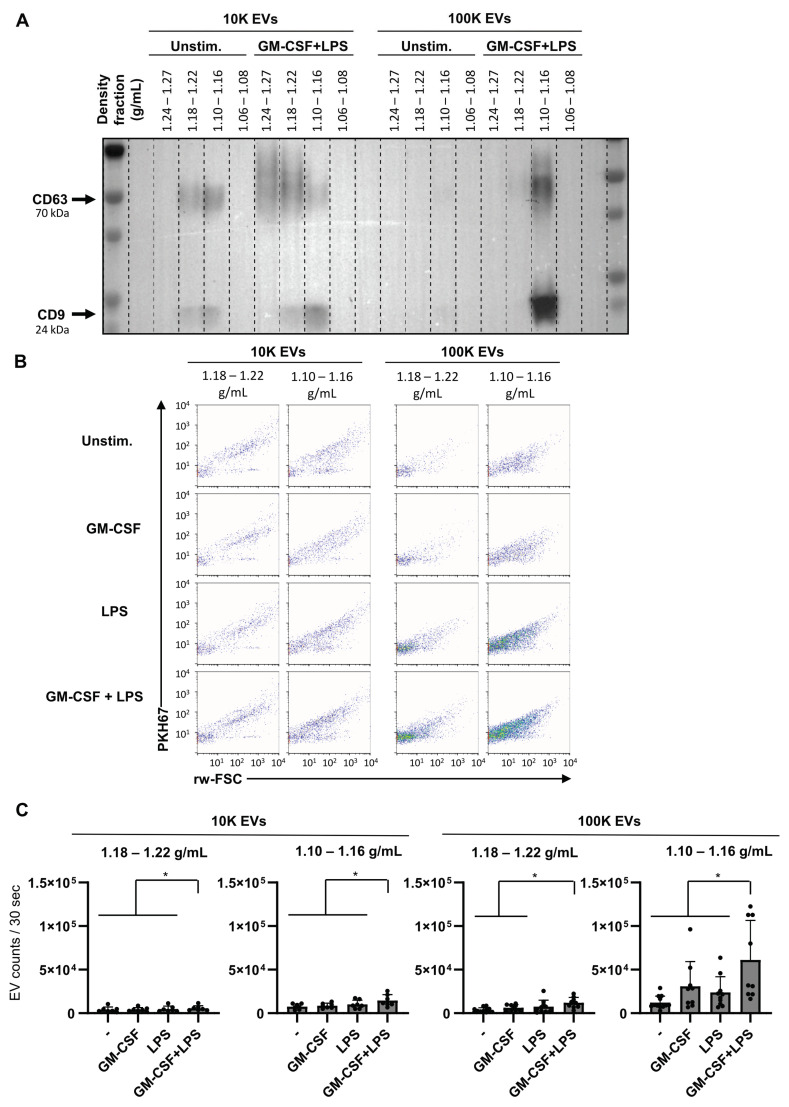
Dual stimulation of neutrophils increases EV release. Neutrophils were cultured for 2 h in the absence of presence of GM-CSF (50 U/mL) and/or LPS (10 ng/mL). After 2 h, culture supernatants were collected for the analysis of EV release. (**A**) Protein analysis (Western blotting) of EVs pelleted at 10 kg and 100 kg and floated into a sucrose gradient from neutrophils from a representative donor. Analysis is shown for CD9 and CD63 (tetraspanins; general EV-markers). (**B**) High-resolution flow cytometric analysis of purified PKH67-labeled EVs pelleted at 10 k and 100 k and floated in a sucrose density gradient from neutrophils stimulated with GM-CSF and/or LPS. (**C**) Quantification of EV release in the EV density fractions (1.10–1.16 g/mL and 1.18–1.22 g/mL) as determined by high-resolution flow cytometry. Indicated are the numbers of detected events within the fixed time frame of 30 s, with multiple conditions in each experiment. Data are presented as mean ± SD. Asterisks indicate significant differences: * *p* < 0.05, (*n* = 7−11), one-way ANOVA and mixed-effects analysis with Tukey’s correction.

## Data Availability

The data presented in this study are available on request from the corresponding author.

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
