# Peer review of "Efficient Neutrophil Activation Requires Two Simultaneous Activating Stimuli"

_ijms, 2021, doi:10.3390/ijms221810106_

Round 1

Reviewer 1 Report

  In this interesting article, the authors demonstrated that neutrophils require simultaneous ligation of two different receptors for their efficient activation. Peripheral blood neutrophils were isolated from healthy donors and cultured with various combinations of stimuli including GM-CSF, fMLF, TNF and LPS, and further evaluated for essential functions including degranulation, ROS production, NETosis, phagocytosis and release of mediator and extracellular vesicle. They found that exposure of neutrophils to any combination of stimuli, but not to single stimuli, resulted in significant degranulation and release of mediator/extracellular vesicle, while ROS production was more dependent on the type of stimulation, and phagocytosis was induced to its maximum capacity by single stimuli. Particularly, in their results, NETosis was not induced by any of the used physiological stimuli. Finally, they concluded that full neutrophil activation is tightly regulated and requires activation by two simultaneous stimuli, not a spatiotemporal separation of these stimulations.

   This interesting study has raised a challenge against the current dogma that neutrophil activation is a two-step process, requiring priming to allow them to respond to an activating stimulus and become fully activated.

  Nevertheless, there are some issues needed to be clarified as follows.

  1. In the experimental results, there were large variations in neutrophil degranulation and ROS production between donors, raising a concern regarding the influence of “immunosenescence” in their healthy neutrophil samples. Indeed, functional decline has been observed in neutrophils donated from aged healthy individuals (Exp Gerontol 2018;105:70).
  2. In their NETosis experiments, despite a clear induction by PMA as a positive control, NETs formation was not induced by any of the used physiological stimuli such as LPS or their combinations. A published article demonstrated a significant age-related decline in LPS-induced NET formation, possibly associated with reduced generation of ROS (Aging Cell 2014;13:690). Although the microbiological origin of LPS in this study was not described in the Materials and Methods section, it has been shown that neutrophils are able to discriminate between LPS of different bacterial sources and thereby selectively release NETs (Front Immunol 2016;7:484).
  3. In Introduction and Discussion sections, the authors discussed the pathogenic involvement of neutrophils in rheumatoid arthritis (ref. 4), with their full activation by the pro-inflammatory cytokines like TNF, which is also highly active in arthritis patients. Actually, neutrophils also participate in the pathogenic mechanisms of systemic lupus erythematosus (Front Immunol 2021;12:649693), and there are abundant circulating TNF levels in such victims (Lupus Sci Med 2018;5:e000260). The authors should extend their discussion on the role of neutrophils in the lupus pathogenesis.

Reviewer 2 Report

Mol et al verified ex vivo using isolated from human blood neutrophils the relevance of the requirement of the two-step neutrophil activation process triggered by dual stimulation using different combinations of GM-CSF, fMLF, TNF and LPS, which could eventually clarify the current designation of priming and activating stimuli.

From what I understood, the major aim was to test the neutrophil activation, by measuring degranulation, ROS generation, cell death, phagocytosis and inflammatory cytokine production using techniques such as flow cytometry, microscopy, ELISA and Western Blot.

Consequently, based on obtained results the authors suggest dismissing the naming difference between priming and activating agents, as ligation to two activating receptors is required, instead of the currently proposed two-step priming and activating process. I believe the science is very well conducted with all appropriate controls. I have only minor issues with agreeing with the authors on their interpretations.

Issues:

  1. Please provide more details on the experiments with LPS (catalog number, from which species was it isolated)? Similar information is missing when it comes to experiments with fMLF, GM-CSF and TNF (alpha?) (Page 15, lines 422 - 424).
  2. The authors analyzed cell death (Figure S1B), however no details on % of markers to assess this was not provided. Please provide the details on performed assay (was that NET-osis, apoptosis, pyroptosis?) Have the authors consider the employment of AnnexinV staining in order to distinguish between different cell death types?
  3. Regarding Figure S2, representative dot-plots would add value to the figure.
  4. The authors performed a very complicated flow cytometric analysis using Flow Jo software. Please provide (describe or paste as a separate supplementary figure) the gating strategy used to analyzed results from Figure 1. 5. Please provide more details on methodological ROS analysis (DHR concentration, incubation details: RT or 37oC, washing steps) (page 15, line 427)
  5. Figures presenting flow cytometric results could have better resolution (Figure 1A, 1B, 3A, 5A, 6B. Moreover, please provide the details on the FlowJo software version used (page 16, line 438). As far I am aware, Tree Star is no longer produced by Tree Star Inc.
